# Targeting FTO Suppresses Pancreatic Carcinogenesis via Regulating Stem Cell Maintenance and EMT Pathway

**DOI:** 10.3390/cancers14235919

**Published:** 2022-11-30

**Authors:** Rachana Garg, Laleh Melstrom, Jianjun Chen, Chuan He, Ajay Goel

**Affiliations:** 1Department of Molecular Diagnostics and Experimental Therapeutics, Beckman Research Institute of City of Hope Comprehensive Cancer Center, Monrovia, CA 91010, USA; 2Division of Surgical Oncology, Department of Surgery, Beckman Research Institute of City of Hope Comprehensive Cancer Center, Duarte, CA 91010, USA; 3Department of Systems Biology, Beckman Research Institute of City of Hope, Monrovia, CA 91010, USA; 4Department of Chemistry and Institute for Biophysical Dynamics, The University of Chicago, Chicago, IL 60637, USA; 5Howard Hughes Medical Institute, The University of Chicago, Chicago, IL 60637, USA; 6Medical Scientist Training Program/Committee on Cancer Biology, The University of Chicago, Chicago, IL 60637, USA; 7Department of Biochemistry and Molecular Biology, The University of Chicago, Chicago, IL 60637, USA

**Keywords:** pancreatic cancer, m^6^A, post-transcriptional modification, FTO, cancer stem cells, EMT

## Abstract

**Simple Summary:**

Current treatment strategies for pancreatic cancer (PC) yield poor survival outcomes. It is thus essential to identify factors that orchestrate pancreatic tumorigenesis and contribute to therapeutic resistance. Fat mass and obesity-associated protein (FTO) play a crucial role in demethylating N6-methyladenosine (a known and prevalent modification in eukaryotic RNA). Notably, recent pioneering studies have identified a significant association of FTO overexpression with the cancer progression of various types. Nevertheless, our current understanding of the role of FTO in modulating biological processes relevant to PC is in its infancy. We demonstrate that PC cells expressed higher FTO levels than the normal pancreatic ductal epithelial (HPDE) cells. Furthermore, both lentiviral-mediated genetic and CS1-mediated pharmacological inhibition of FTO impaired PC cell growth, survival, migratory, and invasiveness capabilities. Additionally, FTO loss led to the reversal of EMT traits, impaired tumorsphere formation, and reduced the expression of cancer stem cells (CSC) markers. In agreement, FTO-depleted PC cells displayed impaired tumorigenic capability in a xenograft model. Our study thus demonstrates the functional importance of FTO overexpression in the PC tumorigenicity and maintenance of CSCs via EMT regulation. Therefore, FTO may represent an attractive therapeutic target for PC.

**Abstract:**

N^6^-methyladenosine (m^6^A) is the most prevalent post-transcriptional RNA modification regulating cancer self-renewal. However, despite its functional importance and prognostic implication in tumorigenesis, the relevance of FTO, an m^6^A eraser, in pancreatic cancer (PC) remains elusive. Here, we establish the oncogenic role played by FTO overexpression in PC. FTO is upregulated in PC cells compared to normal human pancreatic ductal epithelial (HPDE) cells. Both RNAi depletion and CS1-mediated pharmacological inhibition of FTO caused a diminution of PC cell proliferation via cell cycle arrest in the G1 phase and p21cip1 and p27kip1 induction. While HPDE cells remain insensitive to CS1 treatment, FTO overexpression confers enhancements in growth, motility, and EMT transition, thereby inculcating tumorigenic properties in HPDE cells. Notably, shRNA-mediated FTO depletion in PC cells impairs their mobility and invasiveness, leading to EMT reversal. Mechanistically, this was associated with impaired tumorsphere formation and reduced expression of CSCs markers. Furthermore, FTO depletion in PC cells weakened their tumor-forming capabilities in nude mice; those tumors had increased apoptosis, decreased proliferation markers, and MET conversion. Collectively, our study demonstrates the functional importance of FTO in PC and the maintenance of CSCs via EMT regulation. Thus, FTO may represent an attractive therapeutic target for PC.

## 1. Introduction

Pancreatic adenocarcinoma (PDAC) is one of the most aggressive and lethal malignancies worldwide, with a five-year survival rate of less than 10% [1]. The intrinsic aggressiveness of the disease, lack of effective therapies, and acquisition of drug-resistant characteristics are some factors that contribute to the notoriously poor prognosis of patients with PDAC [2]. In addition, due to its highly metastatic nature, most PDACs have distant metastases before their initial diagnosis, making most PDAC patients unresectable [2,3]. This clinical problem thus warrants the identification of alternative therapeutic strategies and novel gene candidates that may be targeted to overcome drug resistance and contribute to the successful treatment of patients with metastatic disease.

Recently, stem cells have been reported in several malignancies, including PDAC [4]. These cancer stem cells (CSCs) possess self-renewal properties and have been considered responsible for tumor persistence, metastasis, recurrence, and acquisition of chemoresistance [5,6]. In addition, pancreatic CSCs, first characterized by Li et al., have been an essential determinant of disease aggressiveness, metastasis, and the development of resistance to conventional therapeutic drugs [7,8]. However, the components of signaling networks contributing to the persistence or increase in CSC populations following the acquisition of drug resistance remain largely undetermined.

To date, > 150 different post-transcription RNA modifications have been identified in eukaryotic transcripts. N^6^-methyladenosine (methylation of N^6^ adenosine; m^6^A) has been recognized as the most abundant modification in eukaryotic mRNA [9,10]. The deposition of m^6^A modification is characterized by DRACH sequence (D = G, A, or U; R = G or A; H = A, C or U) predominantly enriched in the 3’ untranslated regions (UTR) near the stop codons and long internal exons [11]. Recent studies highlight the essential role of m^6^A modification in regulating fundamental cellular processes, including tissue development, cell proliferation, migration, invasion, self-renewal, and differentiation [12].

The m^6^A modifications are reversibly moderated by three classes of regulatory proteins: methyltransferase complex (“writers”), demethylase (“erasers”), and reader proteins (“readers”) [13]. “Writers” consist of a multicomponent methyltransferase complex, including methyltransferase-like 3 (METTL3), METTL14, and Wilms’ tumor 1-associating protein (WTAP1), which serve to mediate the methylation modification process of adenylate residues on RNAs. RNA demethylases (“erasers”) subsequently erase these m^6^A marks of mRNA, e.g., fat mass and obesity-associated protein (FTO) and alkylated DNA repair protein alkB homolog 5 (ALKBH5) proteins. These m^6^A methylated mRNAs are then recognized by “readers” (YTHDF1/2) that preferentially bind to RNA at the G [G > A] m^6^ ACU consensus sequence [14]. 

As m^6^A plays an essential role in RNA splicing, translation, stability, and translocation, deviant modifications in m^6^A affect various pathophysiological processes [15]. Dysregulation in m^6^A has been linked with cancer progression in multiple malignancies, including prostate, breast, pancreatic, colorectal, gastric cancer, glioblastoma, hepatocellular carcinoma (HCC), and acute myeloid leukemia (AML) [11,14,16]. We have recently developed a biologically conserved gene expression panel of seven fundamental m^6^A regulators (RNAMethyPro) that can robustly envisage patient survival in multiple human malignancies [17]. Notably, our network-based analysis revealed a functional alliance among various m^6^A regulators, cancer progression, and EMT signature genes, thus highlighting the potential interplay between m^6^A machinery and cancer metastasis. Furthermore, dysregulation in the expression of individual m^6^A regulators may exert tumor-suppressive or oncogenic roles and have varied phenotypic consequences [18,19,20,21].

FTO, the first identified “eraser” of m^6^A, is highly expressed in AML subtypes and inhibits all-trans-retinoic acid-induced leukemia cell differentiation [22,23]. In addition, FTO overexpression has been associated with poor prognosis in breast, thyroid, gastric and endometrial cancer [24,25,26,27]. In addition, upregulation of FTO has been linked with glioblastoma stem cell self-renewal pluripotency and tumor growth [28]. Furthermore, in a Japanese cohort, the FTO gene variant rs9939609 was reported to be associated with pancreatic cancer risk in an obesity-independent mechanism [29]. Given the importance of FTO overexpression in tumorigenesis, deciphering its function, underlying mechanism, and effectors is relevant to ascertaining its extent as a potential therapeutic target. However, our current understanding of the role of FTO in biologically relevant processes driving pancreatic carcinogenesis is still in its infancy.

The present study demonstrates that FTO is highly overexpressed in pancreatic cancer vs. normal cells. Furthermore, using various cellular, biochemical, and genetic approaches, we show that FTO depletion in pancreatic cancer cells led to decreased cell proliferation, migration, invasion, and CSCs growth and maintenance, establishing the translational significance for targeting FTO as a therapeutic candidate in PDAC.

## 2. Materials and Methods

### 2.1. Cell Culture

The human pancreatic cancer cell lines, MiaPaca2, BxPC3, Capan-1, Capan-2, and PANC-1, were obtained from the American Type Culture Collection (ATCC, Manassas, VA, USA). Cells were maintained in the complete medium containing DMEM (Gibco, Carlsbad, CA, USA) supplemented with 10% fetal bovine serum (Gibco, Carlsbad, CA, USA) and 1% penicillin/streptomycin (Gibco, Carlsbad, CA) at 37 °C in 5% CO_2_. All cell lines were tested for mycoplasma regularly and authenticated by a panel of genetic and epigenetic markers. In addition, the normal human pancreatic duct epithelial cell line (H6c7, HPDE) was generously provided by Dr. Kim Orth, University of Texas S Medical Center. HPDE cells were grown in a keratinocyte serum-free medium supplemented with EGF and bovine pituitary extract (Gibco, Carlsbad, CA, USA). 

### 2.2. Stable Knockdown of FTO in Pancreatic Cancer Cells and Overexpression in HPDE Cells

pLKO.1 puro, pLKO.1-shFTO-1 and pLKO.1-shFTO-2 plasmids were received as a generous gift from Dr. Jianjun Chen (Department of Systems Biology, Beckman Research Institute of City of Hope, CA). Lentiviral particles were generated using HEK-293T cells and the ViraPower Packaging System (Invitrogen). For the generation of stable cell lines, MiaPaca-2 and BxPC3 cells were transduced with either FTO-1, FTO-2, or non-target control lentivirus (NTC) and subsequently were subjected to puromycin selection (6 µg/mL) for 7 days. Puromycin-resistant MiaPaca-2 and BxPC3 shFTO clones (shFTOa and shFTOb) were isolated and screened for FTO mRNA and protein expression by qPCR and Western blot. The FTO overexpression plasmid was also kindly provided by Dr. Chen’s laboratory. FTO expression construct was transfected into HPDE cells using lipofectamine 2000 (Invitrogen, Carlsbad, CA, USA), and 48 h after transfection, cells were used in the indicated experiments. For all transfections, an empty pcDNA vector was used as a control. 

### 2.3. Cell Viability 

Per the manufacturer’s instructions, the cell viability was assessed using a CCK-8 assay kit (Dojindo, Kumamoto, Japan). Briefly, cells were seeded in 96-well plates at a density of 3 × 10^3^ cells/well in 100 μL of complete medium and grown for 24–72 h. To study the effect of FTO inhibitor, cells were first seeded in the 96-well plate, and after 24 h, cells were treated with varying concentrations of Bisantrene dihydrochloride (CS1, 0–800 nM, Sigma-Aldrich) for another 24–72 h. Subsequently, CCK-8 solution (10 μL) was added to every well. After 2 h, absorbance was measured at 450 nm using a microplate reader (Molecular Devices, USA). 

### 2.4. Anchorage-Dependent and Anchorage-Independent Colony Formation Assay

As described previously, a colony formation assay was performed to assess anchorage-dependent cell growth [30]. Briefly, cells were seeded into 6-well plates (500 cells/well) and maintained in a complete medium for  1 week. In some cases, cells were treated with an appropriate FTO inhibitor (CS1) concentration for 1 week, and the medium was refreshed every 3 days. The colonies were fixed with methanol and stained with 0.1% crystal violet (ACROS Organics, Gujarat, India). Subsequently, number of stained colonies were counted and photographed. The experiments were performed in triplicate.

### 2.5. Apoptosis and Cell Cycle Analysis

To study the effect of FTO depletion on apoptosis and the cell cycle, 1 × 10^5^ synchronized MiaPaca2 cells (NTC, shFTOa, and shFTOb) were seeded in the MW 6. After 48 h of cell adherence, the percentage of cells in the G0/G1, S, and G2/M phases of the cell cycle and apoptotic cell fraction were determined using a Cell Cycle Assay Kit and Annexin V and Dead Cell Assay Kit (Millipore, Chicago, IL, USA) on a Muse Cell Analyzer (Millipore) as per the manufacturer’s instructions. In addition, the nuclear blebbing caused due to apoptosis was evaluated by Hoechst staining. Briefly, 1 × 10^5^ cells were seeded on the sterile coverslips in MW 12 and allowed to adhere. Cells were then stained with Hoechst-33258 dye (1 mg/mL) for 20 min and fixed using 4% paraformaldehyde for 15 min. The coverslips were then mounted and observed using a confocal microscope. 

### 2.6. Sphere Formation Assay

Tumor sphere formation assay was performed as described previously with some modifications [30]. Briefly, single-cell suspension of the NTC, shFTOa and shFTOb pancreatic cancer cells were seeded in the MW-6 Costar^®^ ultra-low attachment plates (Corning, Corning NY) at a density of 1000 cells/well in serum-free DMEM-F12 medium (Stem Cell Technologies) supplemented with 1% penicillin, 1% streptomycin (Gibco), 1% B27 supplement (Gibco), 10 ng/mL Epidermal Growth Factor (EGF, Sigma-Aldrich), and 10 ng/mL human recombinant basic fibroblast growth factor (bFGF, GIBCO). The cells were then incubated at 37 °C with 5% CO_2_ to allow the formation of spheroids for 7–10 days. The number of spheres formed in each well was calculated. Spheroids were harvested and processed for RNA and protein isolation. The mRNA and protein expression of stem cell markers were analyzed by qPCR or Western blot.

### 2.7. Migration and Invasion Assay

A wound healing assay was performed as described before to assess the cell migration with some modifications [31]. Briefly, the different stable cell lines were seeded in MW-24 as a sub-confluent monolayer. Cells were serum-starved for 12 h, and a scratch was created using a sterile 10 μL tip, thereby creating a wound. The scratch area was carefully washed with PBS, media with 10% FBS was added, and wound closure was monitored for an additional 24 h. Photographs of scratch were taken using an inverted fluorescence microscope (Carl Zeiss Microscopy, NY, USA) at 0 and 24 h. The wound closure depicting the percentage of migration of cells was calculated using Image J software, V 1.8.0, National Institutes of Health, Bethesda, MD, USA. For assessing cell invasion, cells were serum-starved for 24 h and subsequently seeded into the upper chamber of the Matrigel invasion chamber (Corning) with an 8.0 µm PET membrane following the manufacturer’s instructions. The medium with 10% FBS was added to the lower chamber, and the transwell chambers were incubated at 37 °C with 5% CO_2_ for 24 h. After that, cells on the upper chamber side of the membrane were carefully scraped off with a cotton swab, and the invaded cells were fixed with methanol and stained with 0.1% crystal violet and counted. The images were captured using Zeiss AxioCam 702 sCMOS Mono at 10× magnification. In the case of inhibitor treatment, cells pre-treated with CS1 inhibitor for 24 h were employed in either the scratch or invasion assay described above.

### 2.8. RNA Isolation, cDNA Synthesis, and qPCR

Total RNA was isolated from the subconfluent plates using the RNeasy kit (Qiagen, Valencia, CA). One µg of RNA/sample was reverse transcribed using random hexamers and High Capacity cDNA Synthesis Kit (Applied Biosystems) as described before [32]. Quantitative real-time PCR (qPCR) analysis was performed for FTO, E-cadherin, N-cadherin, vimentin, fibronectin, CD44, Nanog, Sox2, ALDH1, and CD133 using SensiFAST™ SYBR^®^ Lo-ROX as per the manufacturer’s instructions (Bioline). β-actin was used as a housekeeping gene. mRNA fold change was calculated by using the 2^-ΔΔCT^ method. The primer sequences are provided in Appendix A and were procured from Integrated DNA Technologies.

### 2.9. Western Blot Analysis

Extraction of total cellular protein and Western blotting was performed as described previously [33]. The following antibodies and dilutions were used: anti-FTO (Abcam ab124892), anti-p-Akt, anti-total-AKT, anti-p-S6-Kinase, anti-CD133, anti-ALDH1, anti-SOX2, anti-Nanog, anti-CD44, anti-E-Cadherin, anti-N-Cadherin, anti-Fibronectin, anti-Vimentin, and caspase 3 (1:1000, Cell Signaling Technology Inc., Danvers, MA, USA, Cat No: 4060, 4691, 9204. 64326, 12035, 3579, 4903, 3570, 3195, 13116, 26836, 46173, 9662, respectively), anti-caspase 9 (Abcam ab32539), and anti-β-actin (1:50,000, Sigma-Aldrich, St. Louis, MO, USA, A5316). Anti-mouse or anti-rabbit secondary antibodies conjugated to horseradish peroxidase (1:5000, Bio-Rad Laboratories, Hercules, CA, USA, 0300-0108P and AAI46P) were used. Chemiluminescence images were obtained using the ChemiDoc-MP Imaging system (BioRad Laboratories, Philadelphia, PA, USA).

### 2.10. In Vivo Tumor Xenograft Study

For the tumorigenesis experiment, NTC, shFTOa, and shFTOb cells (1 × 10^6^) were injected s.c into the flanks of the 6-week-old male athymic nude mice (Jackson laboratories); 10 mice were included in each treatment group. Tumor formation was monitored for 45 days. Tumor volume was determined with caliper measurements and calculated using the formula L1 × L2 × H × 0.5238, where L1 is the long diameter, L2 is the short diameter, and H is the tumor’s height. All animal studies were carried out following the City of Hope IACUC guidelines.

### 2.11. Statistical Analysis

Statistical analysis was performed for three independent experiments using the GraphPad Prism 8 software (GraphPad Software, Inc., La Jolla, CA, USA). All data is represented as the mean ± standard deviation of three independent experiments. The student’s *t*-test was performed to assess the statistical significance between the two groups. *p* < 0.05 or less was considered statistically significant.

## 3. Results

### 3.1. FTO Is Overexpressed in Pancreatic Cancer Cells

To study the role of FTO in pancreatic cancer, we first screened a panel of pancreatic cancer cell lines as they differ in their malignancy nature and state of carcinogenesis. Interestingly, we found that all pancreatic cancer cells (PC) screened express markedly higher FTO mRNA levels than that observed in the “normal” human pancreatic duct epithelial cells (HPDE) (Figure 1A). This correlated with the elevated FTO protein expression in PC cells compared to HPDE cells (Figure 1B). The FTO protein expression was markedly significant in all the PC cells screed when compared to the normal HPDE cells. MiaPaca2 and BxPC3 cells that express the highest level of FTO compared to normal, as well as other PC cells, were further used to conduct the other experiments in this study.

### 3.2. FTO Inhibition Affects Cell Viability, Migration, and Invasiveness of Pancreatic Cancer Cells

Having observed the overexpression of FTO in pancreatic cancer cells, we next intended to determine its role in pancreatic cancer, and hence, we employed a selective pharmacological inhibitor of FTO, CS1, which inhibit FTO’s m^6^A demethylase activity [34]. Notably, FTO inhibition using this pharmacological inhibitor CS1 resulted in a dose- and time-dependent decrease in viability of MiaPaCa2, Panc1, and BxPC3 pancreatic cancer cells (Figure 1C). Furthermore, both MiaPaCa2 and BxPC3 cells showed impaired anchorage-dependent growth, as determined by clonogenic assay, following treatment with the pharmacological inhibitor CS1 (Figure 1D). Emerging evidence linked FTO with cell migration and invasiveness in vitro in gastric cancer cells [21]. Interestingly, we found that migration of MiaPaca2 cells subjected to CS1 treatment was significantly impeded compared with the vehicle-treated control cells, as assessed by a scratch assay (Figure 1E). Notably, CS1 inhibitor treatment decreases the migration of MiaPaca2 cells in a dose-dependent manner. Similar results were obtained in BxPC3 cells. Assessment of the invasive properties revealed identical results, with a higher dose of CS1 displaying the most substantial inhibition of the invasive potential of MiaPaca2 cells (Figure 1F).

### 3.3. Normal HPDE Cells Remain Unaltered following CS1 Treatment

To determine if the effect of FTO inhibition is causally related to the FTO expression levels, we examined the impact of the FTO inhibitor, CS1, on the viability of normal HPDE cells (expresses low FTO levels, Figure 1A) at different time points (24–72 h). Interestingly, normal HPDE cells were insensitive to the CS1 treatment at all-time points and tested doses (Figure 2A). It is important to note that the HPDE cell viability remains unaltered even at the concentrations of CS1 that significantly hindered the viability of Miapaca2 and BxPC3 pancreatic cancer cells (Figure 1C). Furthermore, no significant alteration was observed in the cell cycle profile of HPDE cells following CS1 treatment (Figure 2B). Thus, results suggest that the FTO inhibitor acts explicitly on the pancreatic cancer cells that overexpress FTO and remain insensitive to the normal HPDE cells with low FTO expression (Figure 1).

### 3.4. FTO Overexpression in Normal HPDE Cells Promotes Proliferative and Migratory Phenotype via Modulating EMT Traits 

To obtain an unambiguous answer, we next overexpress FTO in HPDE cells using the lentiviral approach, which led to increased FTO mRNA and protein expression in normal cells (Figure 2C). Notably, FTO overexpression in normal HPDE cells confers enhancements in cell viability (Figure 2D) and cell growth as determined by the CCK8 and colony formation assay. Furthermore, a significant enhancement in the number of colonies was observed in FTO-overexpressing HPDE cells vs. parental cells (Figure 2E). In addition, we found that migration of FTO-overexpressing HPDE cells was significantly higher than the parental HPDE cells, assessed using a transwell chamber (Figure 2F). We next intended to analyze if the effect of FTO overexpression on the migratory capability of HPDE cells correlates with the epithelial-to-mesenchymal transition. Remarkably, FTO overexpressing HPDE showed downregulation of epithelial marker E-cadherin with a concomitant upregulation of mesenchymal marker N-cadherin, evident at both the mRNA and protein levels (Figure 2G), indicating that FTO overexpression is central for acquiring EMT characteristics.

### 3.5. Stable FTO Depletion in Pancreatic Cancer Cells Impairs Their Cell Growth and Proliferative Capabilities 

We used an RNAi-silencing approach to establish the role of FTO in growth and tumorigenesis. First, MiaPaca2 and BxPC3 cells were stably transduced with shRNA lentiviruses for either FTO or non-target control (NTC). As seen in Figure 3A,B, significant depletion of FTO at both the mRNA and protein levels was achieved using the two shRNA lentiviral sequences in MiaPaCa2 and BxPC3 cells compared to NTC transduced counterparts. Two different sequences (shF1 and shF2) were used in all cases to minimize off-target effects. Expression of FTO was reduced by more than 75% by either sequence. Notably, FTO-depleted MiaPaca2, and BxPC3 cells proliferate significantly slower than the corresponding control cell lines (Figure 3C). Next, we assessed the effect of FTO depletion on PC cell growth. Assays of colony formation in liquid and the semisolid medium revealed that both anchorage-independent and anchorage-dependent growth were impaired in FTO-depleted MiaPaCa2 and BxPC3 cells relative to control cells (Figure 3D,E). These data suggest that FTO plays an essential role in pancreatic cancer cell growth and proliferation.

### 3.6. Loss of FTO Alters the Cell Cycle Profile and Induces Apoptosis in Pancreatic Cancer Cells

As both the pharmacological inhibition of FTO (CS1) and stable depletion of FTO by lentiviral approach impaired the cell proliferation of pancreatic cancer cells, we sought to determine if these effects on cell proliferation are due to perturbation in the distribution of cells in various phases of the cell cycle. As seen in Figure 4A, CS1 caused a cell increment in the G0/G1 phase, with a concomitant decrease in S-phase. Furthermore, this was associated with an increased protein expression of CDK inhibitors: p21^cip1^ and p27^kip1^ (Figure 4B). In addition, stable depletion of FTO led to induction of apoptosis in pancreatic cancer cells. An early apoptotic cell population increment from 18.1% in NTC to 51–65.3% occurred in the FTO-deleted cells (Figure 4C).

Two other experiments showed similar results. Original Western blots are available as Appendix A. Moreover, increased protein expression of key apoptotic markers, caspase 3 and caspase 9, were observed upon FTO loss in contrast to the parental or NTC cells (Figure 4D). Furthermore, the Hoechst staining study showed increased nuclear blebbing, chromatin condensation, and apoptotic cells in the cells with stable depletion of FTO. At the same time, the nuclei of NTC-transduced cells were found to be spherical, and the DNA evenly distributed (Figure 4E). Altogether, our results indicate that FTO regulates cell cycle profile via suppression of p21^cip1^ and p27^kip1^ and that its loss triggers an apoptotic event in pancreatic cancer cells.

### 3.7. FTO Is Required for the Migratory and Invasive Aptitude of Pancreatic Cancer Cells

To assess the involvement of FTO on the migratory ability of PC cells in response to FBS, we first examined the effect of FTO depletion on the percentage of wound closure. Notably, the rate of wound closure in the MiaPaca2 NTC cells was higher than that observed in the FTO-depleted cells. Thus, FTO is required to confer migratory potential (Figure 5A). Assessment of the invasive properties revealed similar results, with MiaPaca2-NTC cells displaying the highest invasiveness compared to the FTO-depleted MiaPaca2 cells.

Both the lentivirus shRNA sequences gave comparable results (Figure 5B). To further explore the mechanism underlying the inhibitory effect of FTO depletion on cell mobility and invasiveness, we analyzed the EMT markers’ protein and mRNA expression (Figure 5C,D). Notably, FTO-depleted cells show an increment in the expression levels of epithelial markers, including E-cadherin. In contrast, there was a concomitant decrease in mRNA and protein levels of mesenchymal markers, such as N-cadherin, vimentin, and fibronectin. Thus, FTO loss mediates the reversal of EMT and hence impairs PC cell motility and invasiveness.

### 3.8. FTO Is Required for the Maintenance of Stemness Traits and Self-Renewal Potential of Pancreatic CSCs

Cancer cells that lose epithelial polarity and acquire mesenchymal features develop invasive and stem cell-like characteristics; these CSCs overture metastases. Hence, having observed the inhibitory effect of FTO depletion on migratory, invasive, and EMT reversal in PC cells, we next sought to assess the impact on their stem-cell physiognomies using varied approaches. First, we performed sphere-formation assays using PC cells with stable FTO depletion to determine if FTO is required to maintain stem cell-like properties. FTO depletion in PC cells significantly suppressed the spheroid formation in MiaPaca2 and BxPC3 cells (Figure 5E). Second, we identified that the loss of FTO dramatically impairs the mRNA and protein expression of CSC markers, including CD44, ALDH1, and SOX2, Nanog, and CD133 (Figure 5F,G), thus indicating the essential role of FTO in the enrichment of the CSC subpopulation. Third, replating of single-cells derived from the spheroids demonstrated that the FTO depletion markedly hampered the formation of the secondary spheres (Appendix A) and suppressed the expression of CSC markers (Appendix A), suggesting the critical role of FTO in the self-renewal capacity of pancreatic CSCs. Overall, our results indicate that FTO is necessary for spheroid formation, maintenance of stem cell marker expression, and the self-renewal potential of pancreatic CSCs.

### 3.9. FTO Depletion Inhibits Pancreatic Cancer Cell Xenograft Growth

To assess the effect of FTO depletion on the tumorigenic potential of PC cells in vivo, we injected MiaPaca2 cells stably expressing either F1 shRNA, F2 shRNA, or NTC shRNA s.c. into the left flank of thymic nude mice (Figure 6A). Inoculation of control MiaPAca2 cells led to tumors with a latency of 12 days. Notably, the commencement of tumor formation in the FTO-depleted cells was remarkably delayed (Figure 6B), and this was further reflected in substantially reduced tumor volume and weight compared with the xenografts with NTC or parental PC cells (Figure 6C). Furthermore, a representative photograph of tumors from different treatment groups (Figure 6D) indicates a significant effect on tumor formation concerning size, volume, and weight upon FTO depletion. In addition, the Western blot analysis of total protein lysates prepared from the tumors from different groups showed decreased proliferation. At the same time, there was a marked induction of cell death in FTO-depleted cells, as evidenced by increased caspase 3 and 9 and a decrease in expression of proliferation marker PCNA. (Figure 6E). Furthermore, protein expression analysis of EMT markers confirmed the reversal of EMT by FTO loss as tumors from xenografts with FTO-depleted cell implantation had upregulated E-cadherin and downregulation of mesenchymal markers N-cadherin and vimentin (Figure 6F). Overall, these results support our in vitro findings demonstrating the essential role of FTO in regulating PC cell proliferation, motility, invasiveness, EMT traits, and stem cell characteristics.

## 4. Discussion

N6-methyladenosine (m^6^A), a dynamic and reversible chemical change in RNA, has added a new escalation in epitranscriptomic RNA modification investigations in recent years [35,36]. Pioneering studies highlight that deviant expression of m^6^A regulators controls cancer self-renewal and cell fate and hence contributes to tumorigenesis [13,37]. Studies from our laboratory and others have shown that m^6^A-related mRNA signature effectively predicts the prognosis of cancer patients in various malignancies [17,38]. Furthermore, a high frequency of copy number variation of m^6^A regulatory genes has been associated with pathologic stage and resected tumor size in various cancer patients [16,39]. Although the involvement of m^6^A regulators in carcinogenesis has been recognized over recent years, the specific roles of individual m^6^A machinery (reader, writer, eraser) in tumor pathogenesis and progression remain only partially understood. FTO, the first identified “eraser” of m^6^A, has been reported to be highly expressed in acute myeloid leukemia (AML) subtypes and to promote AML development [23]. Notably, FTO overexpression is associated with poor prognosis in breast, thyroid, gastric, and endometrial cancer [24,25,26,27]. Likewise, FTO overexpression in cervical squamous cell carcinoma enhanced chemo-radiotherapy resistance via regulating b-catenin expression [40]. Interestingly, FTO knockdown in PC cell lines led to reduced cell proliferation and decreased DNA synthesis [41]. In contrast, certain studies have reported a tumor suppressor role of FTO in breast, colorectal, lung, bladder, and prostate cancer [42,43,44,45,46].

Nevertheless, the role of FTO in pancreatic carcinogenesis remains underexplored to date. In the present study, through a series of biochemical and molecular analyses, we provide evidence demonstrating the essential role of FTO in the development of highly aggressive attributes of pancreatic cancer and, more significantly, in the maintenance of stemness of pancreatic CSCs. Our study provides a novel proof of concept for the role of FTO as a therapeutic target for pancreatic cancer.

In concordance with the existing studies, we found the upregulation of FTO in various pancreatic cancer cells compared to the normal human pancreatic ductal epithelial cells (HPDE) [14,39]. Additionally, using a series of gain- and loss-of-function studies, we unambiguously showed the essential requirement of FTO overexpression in conferring enhanced proliferative advantage to the cancer cells against the normal cells. Utilizing a highly selective FTO inhibitor, CS1, we demonstrated the critical impact of FTO inhibition on the cell viability of pancreatic cancer cells and their cell proliferative, motility, and invasive physiognomies. The observation further strengthens the relevance of our study that normal HPDE cells remain insensitive toward this selective inhibitor of FTO, even at the most prolonged dose duration or at the highest concentration included in the study. These findings provide the rationale for implementing CS1 as a therapeutic candidate for pancreatic cancer. Earlier studies have suggested a link between FTO inhibition and suppression of the proliferation in vitro and in vivo [47,48].

Previous studies with FTO pharmacological inhibitors prompted us to reason that this m^6^A eraser may be highly relevant for pancreatic carcinogenesis, and hence we adopted a second strategy to deplete FTO expression in PC cells using the lentiviral approach. Notably, shRNA-mediated perturbation of FTO in pancreatic cancer cells led to alterations in multiple biologically relevant processes associated with pancreatic carcinogenesis, for example, loss of cell growth, motility, and invasiveness. In addition, we uncovered the critical role of FTO in pancreatic cancer cell proliferation and demonstrated that knockdown of FTO led to cell cycle arrest in the G1 phase, and increment in the expression of cyclin/CDK inhibitors, p21cip1 and p27Kip1, which in turn was reflected in their impaired ability to survive and grow as a tumor in a xenograft mice model. Notably, the observation that loss of FTO is sufficient to inhibit tumor growth of pancreatic cancer cells agrees with the reversal of EMT traits observed upon FTO depletion both in vitro and in vivo in tumors from the xenograft model. Notably, in numerous studies, EMT activation has been linked with the generation of cancer stem cells (CSCs), and a strong association between EMT, stemness, and tumor cells’ metastatic initiating potential has been demonstrated [49].

Pioneering studies have shown that tumor cells with the propensity for invasion display loss of surface E-cadherin and upregulation of N-cadherin and vimentin, the typical characteristics of EMT phenotype [49,50]. The results presented in this manuscript highlight for the first time the reversal of EMT in pancreatic cancer cells following FTO inhibition and, therefore, their responsibility for inhibiting pancreatic cancer progression. Further relevance of our findings is strengthened by the demonstration in low-FTO expressing normal HPDE cells, wherein ectopic restoration of FTO expression confers potential oncogenic phenotype, scrutinized in terms of enhanced proliferative and migratory advantage and conversion of epithelial-to-mesenchymal features, a significant phenomenon associated with tumor initiation and metastasis [50].

Recent studies have highlighted the communal molecular tendencies of CSCs and cells undergoing EMT [51]. Furthermore, EMT has been inveterate as a major player in tumor metastasis and recurrence, an essential function of CSCs [52]. Furthermore, a recent study in AML has demonstrated FTO inhibition’s relevance in curbing the self-renewal of cancer stem cells and immune evasion [34]. However, if FTO has any role in regulating CSC in pancreatic cancer, a significant impediment to successful PDAC treatment remains unknown thus far. In corroboration with the reversal of EMT following FTO loss in pancreatic cancer cells, our study additionally presents here the first evidence indicating that FTO depletion not only led to a decrease in the number of sphere formations (a measure of self-renewal capacity of CSCs) but also demonstrated a marked decrease in the expression of various CSC markers. Hence, we explicitly show FTO’s biological and functional significance in governing PC tumorigenesis and the maintenance of CSC via EMT regulation.

## 5. Conclusions

This study explicitly provides evidence for the overexpression of FTO in pancreatic cancer vs. normal cells. In addition, FTO depletion in PC cells impairs their proliferative, migratory, and invasive aptitude. FTO is necessary for CSC sphere formation, maintenance, and self-renewal. In summary, through a comprehensive mechanistic analysis, our study conclusively presents the first evidence demonstrating the crucial role of FTO in pancreatic carcinogenesis via moderating the cell cycle, EMT processes, and maintenance of CSCs. Furthermore, our results advocate that FTO is a druggable target, and inhibiting FTO overexpression holds significant potential for pancreatic cancer treatment.

## Figures and Tables

**Figure 1 cancers-14-05919-f001:**
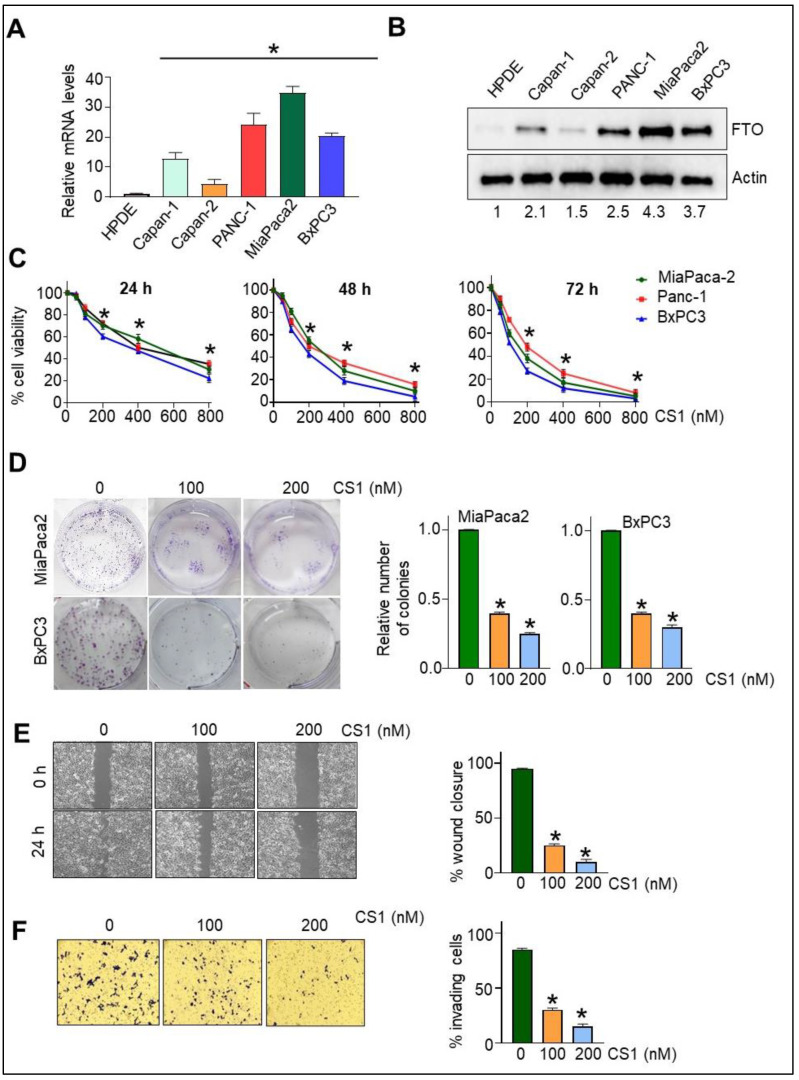
FTO is overexpressed in pancreatic cancer (PC) cells, and its inhibition hampers the growth, migration, and invasion of PC cells. (**A**) FTO mRNA levels as analyzed by qPCR. Data are normalized to actin and expressed as fold increase relative to HPDE cells (n = 3). * *p* < 0.05. (**B**) FTO protein expression as analyzed by Western blot in normal human pancreatic ductal epithelial cells (HPDE) and pancreatic cancer (PC) cell lines. A representative Western blot is shown here. (**C**) MiaPaca2, Panc1, and BxPC3 PC cells were treated with different concentrations of FTO inhibitor, CS1, for different time durations (24, 48, or 72 h), and cell viability was determined by CCK-8 assay. (n = 3). * *p* < 0.05. (**D**) MiaPaca2 and BxPC3 cells (10^3^) seeded in MW6 were pre-treated with different concentrations of CS1, replacing the medium twice a week for 15 days, when colonies were stained with crystal violet and counted. Left: picture shows the representative photomicrograph from the clonogenic assay. Right: relative quantification of colonies per well. (**E**) In the scratch assay, MiaPaca2 cells pre-treated with different concentrations of CS1 were seeded in MW24. Wound closure in response to 5% FBS as determined after quantitative determination of % wound closure, expressed as mean ± SD of triplicate measurements. Two experiments gave similar results. * *p* < 0.05. Similar results were obtained in BxPC3 cells. (**F**) Invasion in the transwell chamber with Matrigel in response to FBS (5%) was determined 16 h after seeding. Left: representative images. Right: quantification of invading cells by contrast microscopy in five independent fields. Results are expressed as mean ± SD of triplicate measurements conducted in MiaPaca2 cells. Two experiments gave similar results. * *p* < 0.05. Similar results were obtained in BxPC3 cells. Original Western blots are available as Appendix A.

**Figure 2 cancers-14-05919-f002:**
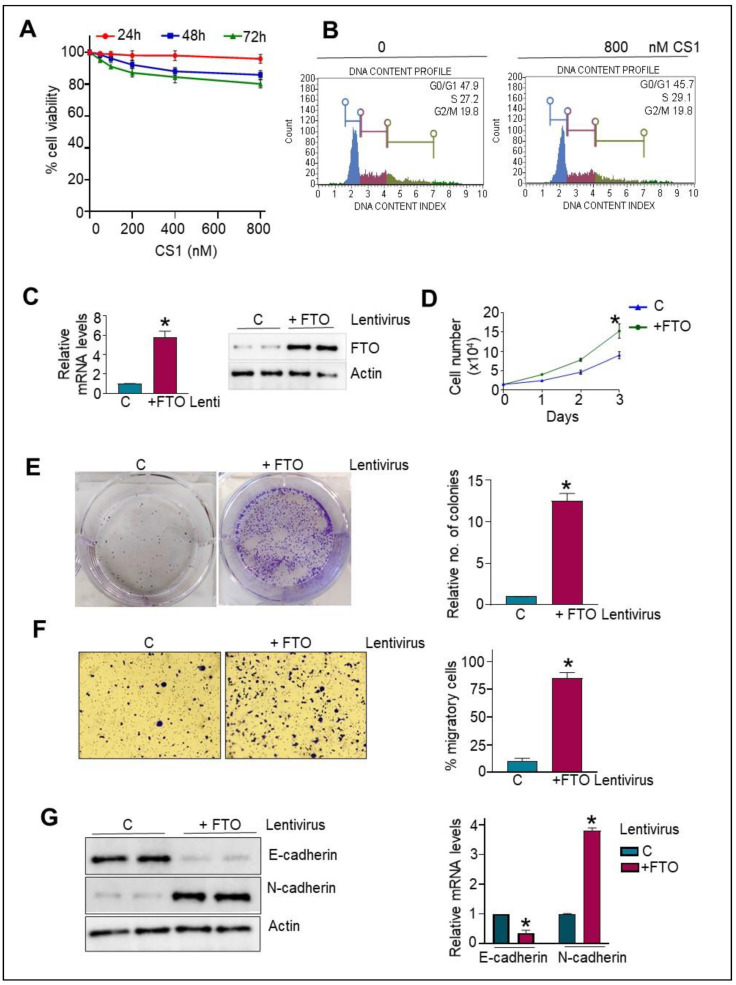
Normal HPDE cells remain insensitive to FTO inhibition, while overexpression confers proliferative and migratory advantage. (**A**) Normal HPDE cells were treated with different concentrations of CS1 for different time points (24, 48, and 72 h), and the CCK-8 assay determined the effect of CS1 on the cell viability. (**B**) The effect of CS1 treatment on the distribution of HPDE cells in various phases of the cell cycle. (**C**) Normal HPDE cells were transduced with either FTO overexpressing or control lentivirus. FTO mRNA levels (left) and protein expression (right) were analyzed by qPCR and Western blot. (**D**) Control (c) and FTO-overexpressing HPDE cells (5 × 10^3^) were seeded in 96-well plates, and cell growth was determined by CCK-8 assay at different time intervals (24, 48, and 72 h). * *p* < 0.05. (**E**) Control (c) and FTO-overexpressing HPDE cells (10^3^) were seeded in MW6, the medium was replaced twice a week, and after 15 days, colonies were stained with crystal violet and counted. Left: Representative photomicrograph from the clonogenic assay. Right: relative quantification of colonies per well. n = 3; * *p* < 0.05. (**F**) Migration in the transwell chamber in response to FBS (5%) was determined 16 h after seeding. Left: representative images. Right: quantification of invading cells by contrast microscopy in five independent fields. Results are expressed as mean ± SD of triplicate measurements. Two experiments gave similar results. * *p* < 0.05. (**G**) Protein and mRNA expression of markers of EMT were determined by Western blot using the total cell extracts and qPCR from the mRNA isolated from the control or FTO-overexpressing HPDE cells. Two experiments gave similar results. * *p* < 0.05. Original Western blots are available as Appendix A.

**Figure 3 cancers-14-05919-f003:**
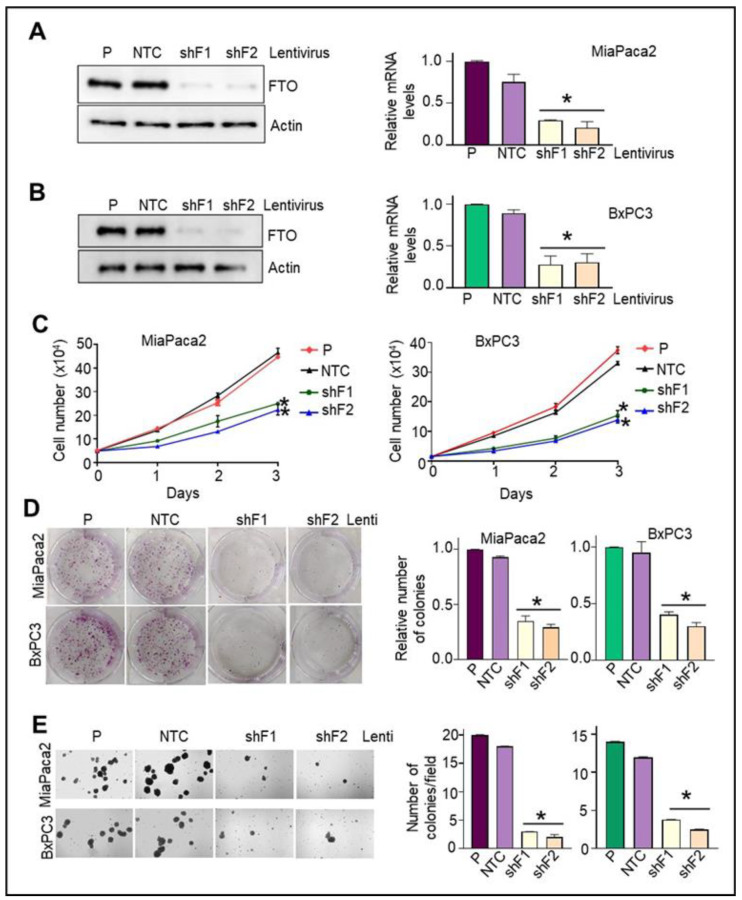
FTO is required for PC cell growth, migration, and invasion. (**A**) FTO stable depletion in MiaPaca2 and BxPC3 was achieved using two different FTO lentiviruses. Control cells were infected with non-target control (NTC) lentivirus. This was followed by selection with puromycin. FTO mRNA and protein expression in following knockdown FTO in PC cells was determined by Western blot and qPCR, respectively. (**B**) MiaPaca2 and (**C**) BxPC3 cells (5 × 10^3^) were seeded in 96-well plates, and allowed to grow in 2% fetal bovine serum. Cell growth was determined by CCK-8 assay at different time intervals (24, 48, and 72 h). * *p* < 0.05. (**D**) MiaPaca2 and BxPC3 cells (103) were seeded in MW6, the medium was replaced twice a week, and after 15 days, colonies were stained with crystal violet and counted. Left: Representative photomicrograph is shown. Right: relative quantification of colonies per well. n = 3; * *p* < 0.05. (**E**) To evaluate anchorage-independent growth, 3 × 10^3^ cells were plated in 0.35% agar over a 0.5% agar layer. After 10 days, the plates were stained with 3-(4,5-dimethylthiazol-2-yl)-5-(3-carboxymethoxyphenyl)-2-(4-sulfophenyl)-2H-tetrazolium (MTS). The number of colonies was counted in five different fields and averaged for each well. Data are expressed as mean ± SD of triplicate measurements. * *p* < 0.05. Original Western blots are available as Appendix A.

**Figure 4 cancers-14-05919-f004:**
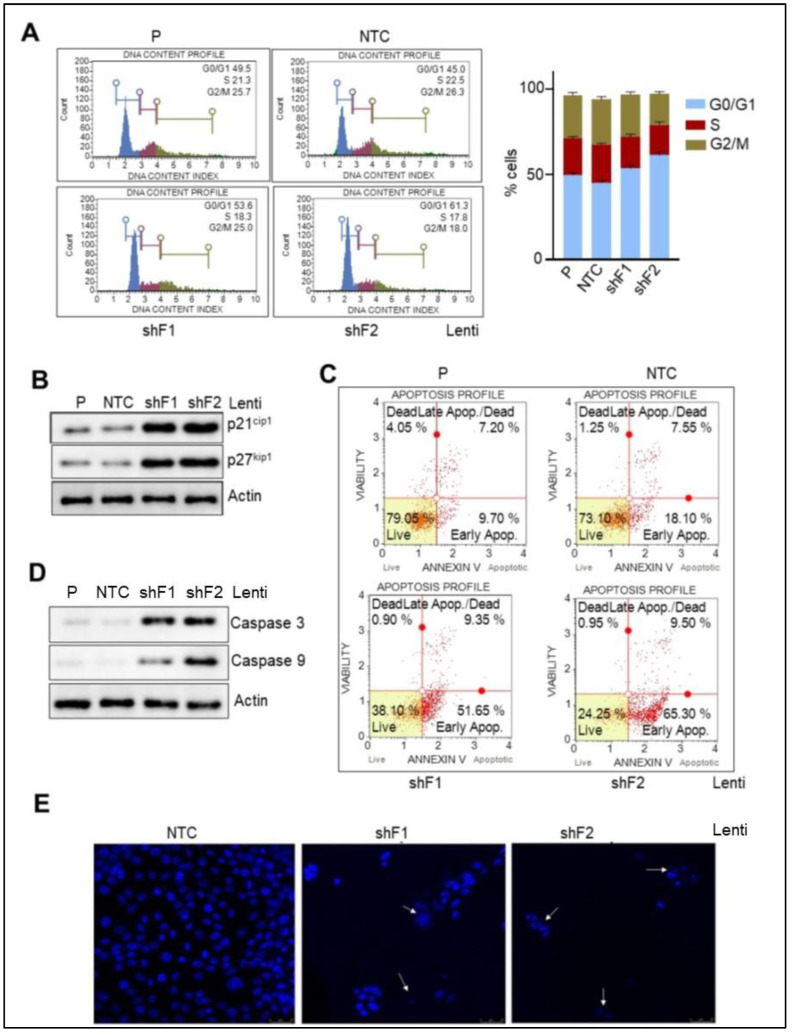
FTO depletion controls cell cycle progression while it induces apoptosis in PC cells. (**A**) NTC and the two FTO-depleted stable PC cells were analyzed for the cell cycle using a Muse Cell Cycle Assay Kit. Left: Representative images depicting the distribution of cells in different phases. Right: Quantitative determination of cell cycle profile. Two experiments gave similar results. (**B**) The Western blot analyzed the protein expression of the CDK inhibitors in the protein extracts from the NTC and the two FTO-depleted stable PC cells. Representative blot images are shown. (**C**) Representative pictures of apoptosis induction in the NTC and the two FTO-depleted stable PC cells as analyzed by annexin V5 assay. (**D**) The Western blot determined the protein expression of the apoptotic markers. Representative images are shown. (**E**) Confocal images representing apoptosis of cells resulting in nuclear blebbing of MiaPaca2 cells following FTO loss. White arrows indicate nuclear blebbing in apoptotic cells (×200 magnification).

**Figure 5 cancers-14-05919-f005:**
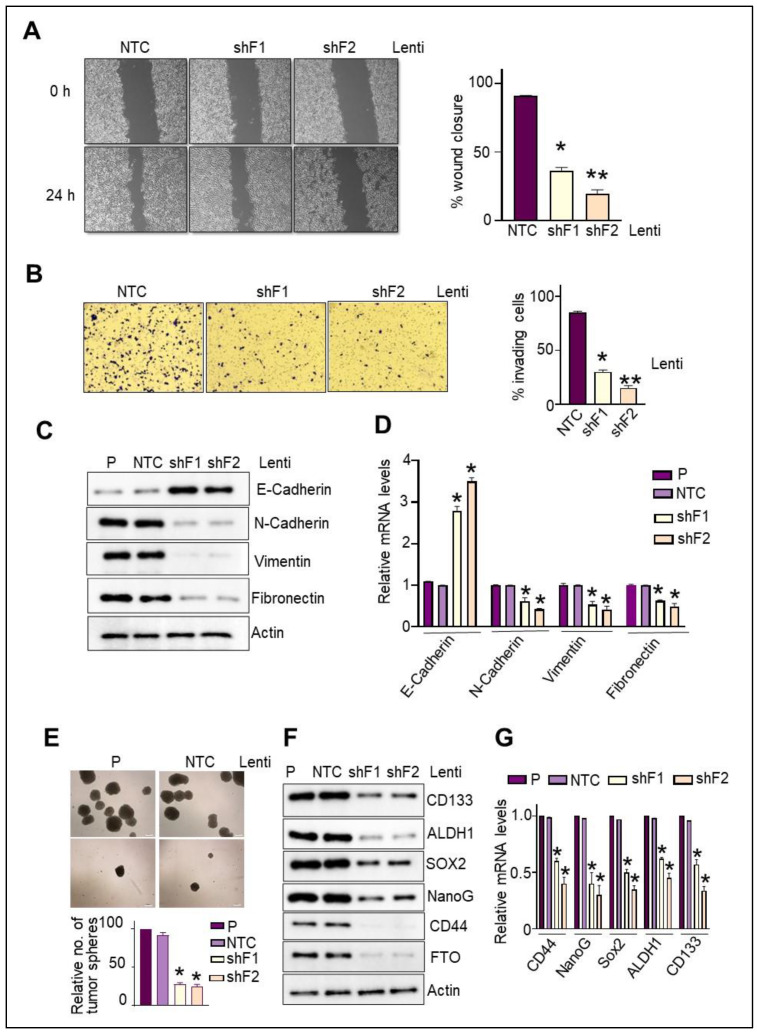
FTO depletion impairs PC cell migratory, invasive, EMT, and stemness characteristics. (**A**) NTC and the two FTO-depleted, stable PC cells were seeded in MW24, and serum was starved for 16 h. Wound closure in response to 5% FBS was determined after making a scratch on the confluent monolayer of cells. Left: Representative images. Right: Quantitative determination of % wound closure, expressed as mean ± SD of triplicate measurements. Two experiments gave similar results. * *p* < 0.05, ** *p* < 0.01. (**B**) Invasion in the transwell chamber with Matrigel in response to FBS (5%) was determined 16 h after seeding. Left: representative images. Right: quantification of invading cells by contrast microscopy in five independent fields. Results are expressed as mean ± SD of triplicate measurements. Two experiments gave similar results. * *p* < 0.05. (**C**,**D**) Protein and mRNA expression of various EMT markers were determined by Western blot in the total cell extracts and qPCR using the mRNA isolated from the NTC and the two FTO-depleted stable MiaPaca2 cells. Two experiments gave similar results. * *p* < 0.05. (**E**) Representative images and relative quantification of the spheres formed in the NTC and the two FTO-depleted stable MiaPaca2 cells. Two experiments gave similar results. * *p* < 0.05. Scale bar: 200 μm (**F**,**G**) Protein and mRNA expression of various stem-cell markers were determined by Western blot in the total cell extracts and qPCR using the mRNA isolated from the spheroids formed in various groups, including NTC and the two FTO-depleted stable MiaPaca2 cells. Original Western blots are available as Appendix A.

**Figure 6 cancers-14-05919-f006:**
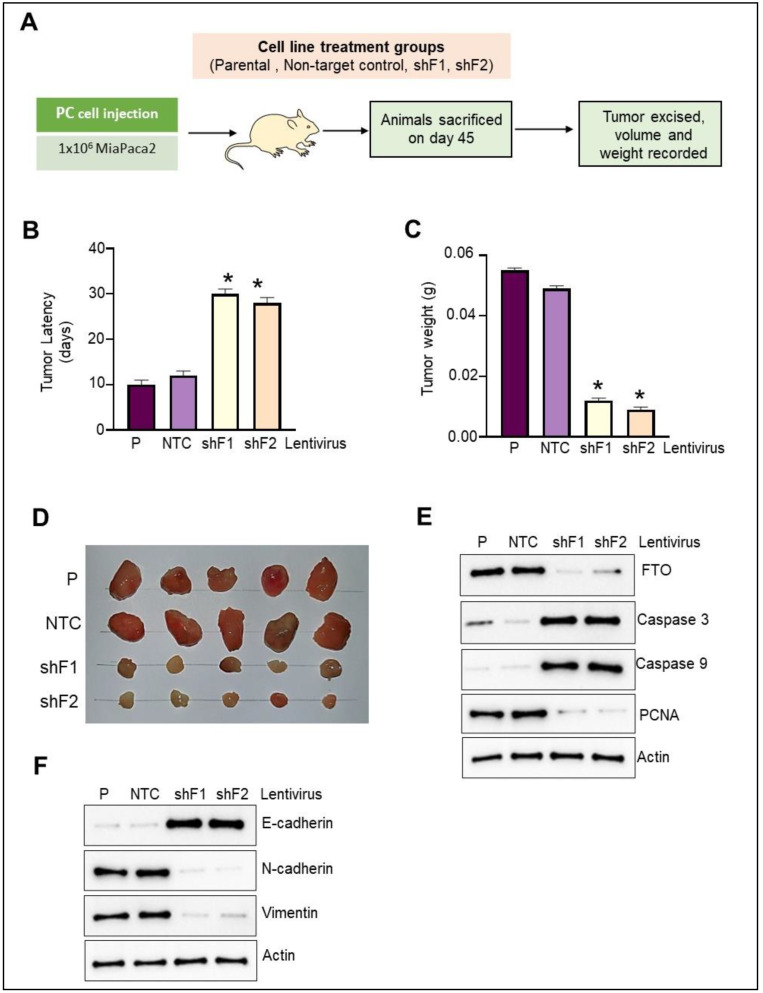
FTO is required for PC tumor growth in athymic nude mice. MiaPaca2 cells (parental, expressing shRNA control (NTC) or FTO (shF1 and shF2) at 80% confluency were resuspended in serum-free medium, and then 0.1 mL containing 1 × 10^6^ cells were injected subcutaneously into the flank of 6-week old male athymic nude-Foxn1nu mice. (**A**) Experimental design. (**B**) FTO depletion caused the delay in the tumor onset, which is depicted as the tumor latency. Each group includes 10 mice. (**C**) Graph representing final tumor weight and (**D**) a representative image of the xenograft tumors. (**E**) protein expression of apoptosis and proliferation markers as analyzed by Western blot in the total cell extracts prepared from the xenograft tumors from different groups. Representative blots are shown. (**F**) Protein expression of the EMT markers as analyzed by Western blot in the total cell extracts prepared from the xenograft tumors belonging to different groups. Representative blots are shown. Two experiments gave similar results. * *p* < 0.05. Original Western blots are available as Appendix A.

## Data Availability

Data are contained within the article.

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
