# Peer review of "Targeting FTO Suppresses Pancreatic Carcinogenesis via Regulating Stem Cell Maintenance and EMT Pathway"

_cancers, 2022, doi:10.3390/cancers14235919_

Round 1

Reviewer 1 Report

In this work entitled “Targeting FTO suppresses pancreatic carcinogenesis via regulating stem cell maintenance and EMT pathway” Garg and collaborators studied the role of FTO in pancreatic cancer. Employing in vitro and in vivo models the authors highlighted the relevance of FTO in the stemness phenotype of pancreatic cancer. Although the work is interesting, several concerns must be addressed for its publication in Cancers.

- Materials and methods section is well organized, however, some relevant information is missing. For example, authors must clarify what is NTC and include the catalogue number of the antibodies used for western blot.

- Regarding cell culture, five different cell lines are mentioned. The authors must explained why different cell lines are employing in different experiments.

- Please, improve the paragraph of Cell viability. The meaning of CS1 is explained several times and there are duplicate sentences.

- In wound healing assay photographs were taken at 0 and 24 h. Did the authors evaluate intermediate times? Do not they think that a shorter time is better to evaluate migration capacity? Please, give a suitable explanation.

- Information about number of animals employed, sex, age, etc must be added in 2.10 section.

- In figure 1A there is only one asterisk, however, it is not clear which cell lines present significant differences respect to the control cell line (HPDE). This problem should be fixed.

- In the legend of figure 1B the authors must include that “representative” bands are shown. Additionally, I suggest to include the densitometric quantification of the three replicates, and include a bar graph and the statistical analysis.

- Statistical analysis must be incorporate in Figure 1C.

- In the legend of Figure 1D (and 2E) it must be mentioned that a clonogenic assay was performed. Why did authors only include MiaPaCa2 and BxPC3 cells and exclude Panc-1? Why wound healing assays were only performed in MiaPaCa2 cells? Which cell line was used in Figure 1F? It is not mentioned in the legend.

- In the legend of Figure 2A is missing the 48 h time point.

- The sentences “two experiments gave similar results” (line 305, 368, etc) and “Two other experiments showed similar results” (line 375) lack of scientific soundness. Thus, only two replicates were performed in figure 1G? Please, clarify.

- Which cell line was used in Figure 1F? It is not mentioned in the legend.

- In line 352 authors must include that NTC is the abbreviation of non- target control.

- It would be interesting to include the analysis of Ki67 in P, NTC, sh1 and sh2 cells.

- The legend of Figure 4D is incomplete. Please, improve it. The representative bands shown correspond to total Caspase 3 and 9 or to cleaved Caspase 3 and 9? Please, clarify.

- The information concerning magnification is missing in Figure 4D.  Please, also include a scale bar.

- Photographs (with scale bar) of spheres must be included in figure 5E.

- Why MiaPaCa2 cells were selected for in vivo experiments? N=?

- The discussion is well organized, however, the authors should deepen the discussion of their results.

Reviewer 2 Report

The authors explored the FTO expression and function in pancreatic cancers cell lines. FTO is highly evaluated in pancreatic cancer cell lines compared to the normal cell and they found that FTO could  affect cell growth, survival, migratory, and invasiveness capabilities both in vitro and in vivo by manipulating FTO expression.

The figures look good, and the experiment design is rational.. 

However, there are several publications about the FTO role in pancreatic cancer which decrease the novelty of this manuscript.

Oncol Lett. 2019 Feb; 17(2): 2473–2478.; Epigenetics. 2022 Apr 11;1-15 ; Cancer Med. 2022 Jul 25. doi: 10.1002/cam4.5054. et al.

Some other comments:

1.Clarify your difference with other FTO published similarly in other papers.

2.For more clinical relevance, you could investigate some data to verify the FTO expression in pancreatic cancer compared with normal. For example, the TCGA database.

3.FTO plays controversial roles in different cancers, could be disscussed about.

4.Some small errors should be carefully checked, eg Line 221, "in"should be capitalization.
